# The Role of Matrix-Bound Extracellular Vesicles in the Regulation of Endochondral Bone Formation

**DOI:** 10.3390/cells11101619

**Published:** 2022-05-12

**Authors:** Barbara D. Boyan, Niels C. Asmussen, Zhao Lin, Zvi Schwartz

**Affiliations:** 1Department of Biomedical Engineering, College of Engineering, Virginia Commonwealth University, Richmond, VA 23284, USA; zschwartz@vcu.edu; 2Wallace H. Coulter Department of Biomedical Engineering, Georgia Institute of Technology, Atlanta, GA 30332, USA; 3School of Integrated Life Sciences, Virginia Commonwealth University, Richmond, VA 23284, USA; asmussennc@vcu.edu; 4Department of Periodontics, School of Dentistry, Virginia Commonwealth University, Richmond, VA 23284, USA; zlin@vcu.edu; 5Department of Periodontics, University of Texas Health Science Center at San Antonio, San Antonio, TX 78229, USA

**Keywords:** exosomes, matrix vesicles, microRNA, 1α,25(OH)_2_D_3_, 24R,25(OH)_2_D_3_

## Abstract

Matrix vesicles are key players in the development of the growth plate during endochondral bone formation. They are involved in the turnover of the extracellular matrix and its mineralization, as well as being a vehicle for chondrocyte communication and regulation. These extracellular organelles are released by the cells and are anchored to the matrix via integrin binding to collagen. The exact function and makeup of the vesicles are dependent on the zone of the growth plate in which they are produced. Early studies defined their role as sites of initial calcium phosphate deposition based on the presence of crystals on the inner leaflet of the membrane and subsequent identification of enzymes, ion transporters, and phospholipid complexes involved in mineral formation. More recent studies have shown that they contain small RNAs, including microRNAs, that are distinct from the parent cell, raising the hypothesis that they are a distinct subset of exosomes. Matrix vesicles are produced under complex regulatory pathways, which include the action of steroid hormones. Once in the matrix, their maturation is mediated by the action of secreted hormones. How they convey information to cells, either through autocrine or paracrine actions, is now being elucidated.

## 1. Introduction

Matrix vesicles (MVs) are extracellular organelles ranging in size from 50 to 150 nm in diameter that are anchored to the extracellular matrix (ECM) of mineralized tissues via integrin binding to collagen. They were first identified in the 1960s by transmission electron microscopy of calcifying neoplasms, as well as in calcifying tissues, including the mammalian growth plate, the osteoid synthesized during primary bone formation, in dentine during tooth formation, and in the intima of blood vessels undergoing calcification [1,2]. In all cases, they shared a circular morphology and a bilaminar membrane [3,4,5]. Because they were originally identified at sites where the first calcium phosphate crystals were deposited in the matrix [6], the early publications describing them focused on their role in mineralization [7,8,9].

Methods for isolating MVs led to the observation that they were enriched with tissue non-specific alkaline phosphatase (TNAP), an ecto-enzyme that is anchored to the outer leaflet of the MV membrane via glycosylphosphatidylinositol [10] and is also present in the plasma membranes of the parent cells [1]. This observation led to the use of alkaline phosphatase-specific activity as the defining characteristic of matrix vesicles isolated by enzymic digestion of the tissue, followed by differential ultracentrifugation of the ECM digest. Biochemical analysis of the isolated MVs provided additional information that explained how they contribute to calcification, first by providing sites for initial calcium phosphate crystal formation and via enzymes that increase local phosphate ion concentration. Subsequent studies demonstrated that MVs also contained enzymes that could process the ECM, facilitating crystal growth, and that MV composition and function were regulated by systemic hormones, including 1α,25-dyhydroxyvitamin D3 [1α,25(OH)_2_D_3_] [8,11,12,13,14,15].

The ability to isolate MVs from cultured chondrocytes and osteoblasts has enabled investigators to gain a more complete understanding of their composition and the role that they play in regulating the cells that produce them. The goal of this review is to provide a synopsis of these studies, using the mammalian growth plate as the primary model system, given the depth of information that exists in the literature. As our understanding of extracellular vesicles has increased, the question of whether MVs are a subspecies of exosomes is important to address, particularly in light of the finding that MVs contain microRNAs that are also reported to be in exosomes. The mammalian growth plate as a model system is no less valuable in this regard, as it provides us with a well-defined linear system for tracking how MVs change as a function of space and time in vivo, and these relationships are retained in culture.

## 2. The Mammalian Growth Plate

The model system that has yielded the most information about matrix vesicles is the mammalian growth plate. Mammalian bone growth is driven by two types of ossification. Endochondral ossification is responsible for long bone growth, while intramembranous ossification primarily produces the flat bones found in the skull [16]. Both processes stem from mesenchymal tissue.

Embryologically, the skeletal bone begins as a cartilage template. This template forms a region of transition in which the chondrocytes align in columns, termed the growth plate. This process is regulated by a number of factors, including parathyroid-hormone-related peptide (PTHrP) and Indian hedgehog (Ihh) [17,18,19]. As the chondrocytes undergo terminal differentiation, they mineralize their ECM and secrete proteins that recruit osteoclasts to resorb the calcified cartilage as well as vascular endothelial cells, resulting in capillary invasion of the tissue [20]. Osteoprogenitor cells then migrate to the sites via the newly formed vasculature, differentiate into osteoblasts, and form bone [21].

A similar process occurs during post-fetal long bone growth, where the growth plate is located between epiphysis and metaphysis and at the chondro-occipital junction at the base of the skull, as well as at the interface of bone and the costochondral cartilage. All of these sites are characterized by a region containing chondrocytes in a glycosaminoglycan-rich, type II collagen matrix [22,23]. These cells respond to regulatory signals to align in columns and undergo terminal differentiation [20]. Growth plates can be divided into different zones populated by chondrocytes with distinct phenotypical characteristics. Chondrocytes mature as they move through the growth plate, passing from the proliferating cell zone through the prehypertrophic and hypertrophic cell zones and finally into the calcified cartilage zone [20]. The rate and extent of this process are under hormonal control, including thyroid hormone [24], estrogen and testosterone [25,26,27], and the vitamin D3 metabolites, 1α,25(OH)_2_D_3_ and 24R,25-dyhydroxyvitamin D3 [24R,25(OH)_2_D_3_] [28,29,30,31,32].

## 3. Matrix Vesicles: Extracellular Matrix Microsomes

Matrix vesicles are a distinctive feature of the growth plate ECM. They are produced by the growth plate chondrocytes and appear to be bud off laterally from the chondrocyte parent cell, although the exact process by which this occurs remains unclear [5]. Recent work using osteoblasts implicates Stx4a, a soluble N-ethylmaleimide-sensitive factor attachment protein receptor (SNARE) protein, in the release of matrix vesicles into the matrix as Stx4a conditional knockout mice were observed to contain fewer matrix vesicles [33]. Two other types of SNARE proteins (Snap23 and Vamp2) were also found to be differentially expressed though their role in vesicle release was not investigated.

MVs are anchored within the ECM via integrin binding to type II collagen [3]. They can be isolated from growth plate cartilage by digesting the ECM with collagenase and hyaluronidase followed by differential ultracentrifugation and have been shown to be heterogeneous in content based on electrophoretic mobility and biochemically [34,35]. Their payload consists of minerals, enzymes, factors, and microRNAs with distinct packaging dependent on where in the growth plate they are produced [36,37,38]. While much is known about the morphology and composition of these extracellular organelles in their role as initial sites of calcium phosphate formation, we are only now beginning to understand how they contribute to the overall physiology of the growth plate.

Matrix vesicles possess a unique phospholipid composition, with high levels of cardiolipin and sphingomyelin compared to the plasma membrane [39]. Phosphatidylserine and phosphatidylinositol are enriched in the inner leaflet of the phospholipid bilayer [4,40]. Calcium–phosphatidylserine–phosphate annexin complexes, which are found at hydroxyapatite nucleation sites, are present in matrix vesicles [41,42] together with a proteolipid phosphatidylinositol complex capable of mineral deposition in vitro [43]. This suggested that the primary role of the extracellular organelle was to provide sites for initial crystal formation in tissues, such as the hypertrophic cell zone of the growth plate, primary bone, or fracture callus, where there was no pre-existing mineral [1].

Other data supported this hypothesis. When first formed, matrix vesicles contain high levels of magnesium and adenosine trisphosphate (ATP), known inhibitors of calcification [44,45], as well as carbonic anhydrase [46], and ectonucleotide pyrophosphatase/phosphodiesterase 1 (ENPP1) [42,47]. High levels of alkaline phosphatase activity are present along with the mineral octacalcium phosphate. As MVs mature in the matrix, they become leaky due to the action of phospholipases, which are regulated via 1α,25(OH)_2_D_3_ that is secreted by the chondrocytes [48]. Calcium ions are able to diffuse in, and at the same time, the active transport of Ca^++^ out of the vesicles is reduced by the action of ATPases on ATP, generating the calcification inhibitor pyrophosphate and adenosine monophosphate (AMP) [49]. The Ca^++^ eventually dilutes the inhibitory effects of Mg on apatite formation. Ultimately, pyrophosphatase (PPase) acts on pyrophosphate to form free phosphate, which can then interact with Ca^++^ to form apatite crystals on the inner leaflet of the membrane [50]. In addition, MVs possess ion transporters, particularly for phosphate, enabling its enrichment within the microsomes during crystal formation [1,47,51]. Another phosphatase present in matrix vesicles is phosphoethanolamine/phosphocholine phosphatase 1 (PHOSPHO1), which acts on phosphocholine and phosphoethanolamine to initiate crystal formation [47]. As the MV breaks down, these initial crystals can provide sites for epitaxial growth of apatite crystals within the collagen matrix. Matrix vesicle TNAP participates in this process by modulating the phosphorylation status of osteopontin in the matrix, thereby facilitating crystal growth [52] (Figure 1A–E).

A similar process has been studied in osteoblast-derived matrix vesicles, where a group of membrane transporters and enzymes has been demonstrated to be involved in the regulation of hydroxyapatite crystal nucleation and growth. ENPP1 found on the outer leaflet of osteoblast matrix vesicles is involved in regulating hydroxyapatite crystal growth by generating the inhibitor pyrophosphate from ATP, while annexin and ankylin protein (ANK) provide a transmembrane channel for pyrophosphate to enter the extracellular organelles [53]. TNAP, also located on the outer leaflet, is able to degrade the pyrophosphate into monophosphate ions required for crystal growth, while two matrix vesicle membrane transporters, sodium/phosphate co-transporter type III (PiT1) and annexin V, are able to transport monophosphate ions and Ca^++^, respectively, into the vesicles. By impacting the makeup and location of specific ions, these enzymes and transporters can regulate crystal growth.

Matrix vesicles can be isolated from tissues such as growth plate cartilage through a one-step purification process. Their isolation requires initial digestion of the ECM via collagenase and/or hyaluronidase, followed by careful removal of the intact cells via centrifugation and differential centrifugation of the supernatant to pellet large organelles such as mitochondria, nuclei, endoplasmic reticulum, endosomes, and plasma membrane (PM). The remaining supernatant is centrifuged at high speed to pellet the matrix vesicles. Because the MVs are not lysed during isolation, their native conformation remains intact; thus, they remain right side out, with the ectoenzyme TNAP facing outward. In contrast, PMs are isolated from lysed cells and may form liposomes with the outer leaflet of the membrane facing the interior, away from the TNAP substrate in the media. Scientists have taken advantage of this to define MV purity as having alkaline phosphatase specific activity that is a minimum of two-fold greater than that of the PM fraction [54]. In addition, their relative purity can be readily determined using marker enzymes and proteins, including validation that they are not mitochondrial membrane fragments based on their high cardiolipin content [15,39,55].

Using these criteria to identify MVs, investigators have been able to address questions related to MV production, composition, and function. Studies using somatic cell hybrids to monitor alkaline phosphatase enrichment demonstrated that matrix vesicles are produced as specific organelles and are not simply membrane debris left in the matrix during ECM synthesis [56].

This same approach can be used to isolate matrix vesicles from cell cultures, which has enabled careful cataloging of their contents and how they are regulated. Studies from a variety of laboratories have shown that they possess annexins 2, 5, and 6 [51], as well as various enzymes that process the ECM, including matrix metalloproteinases (MMP2, MMP3) [36,48], a disintegrin and metalloproteinase with thrombospondin motifs (ADAMTS), and tissue inhibitor of metalloproteinases (TIMP1, TIMP2). Given their size, 50–150 nm diameter, and the complexity of their composition, it is likely that there is more than one kind of matrix vesicle. Indeed, differential centrifugation of isolated matrix vesicles shows that there are different populations that vary in density as well as alkaline phosphatase content [57].

Matrix vesicles have been isolated from soft tissues as well as mineralizing tissues. They are present in vascular smooth muscle [58], and their content changes at vascular sites that are becoming mineralized [42]. Similarly, there are distinct differences in matrix vesicles isolated from the upper growth plate, which does not mineralize, and those isolated from hypertrophic cartilage, which is becoming calcified during endochondral bone formation. Matrix vesicles are also present in tissues that do not become calcified, including some cancers [1,2]. In general, they all contain enzymes associated with matrix modification.

Studies comparing matrix vesicles produced by resting zone growth plate (RC) cartilage and those produced by prehypertrophic/upper hypertrophic growth zone (GC) cartilage show that they differ markedly [59]. They possess a different membrane phospholipid composition; their alkaline phosphatase activity differs, and their MMP content differs. RC matrix vesicles are enriched in neutral MMPs, whereas GC matrix vesicles are enriched in acid MMPs [36]. This implies that the production of the microsomes is under genomic regulation, but once in the matrix, control of their function must involve non-genomic mechanisms, as they lack the machinery necessary for gene transcription and protein synthesis.

## 4. Regulation of Matrix Vesicle Function

Production of matrix vesicles is under genomic regulation. Not surprisingly, knockout mice lacking expression of key components such as TNAP, PPase, or PHOSPHO1 [50], produce matrix vesicles that lack these proteins and whose function is altered as a result. Direct evidence of genomic regulation by steroid hormones supports this. The vitamin D metabolites 1α,25(OH)_2_D_3_ and 24R,25(OH)_2_D_3_ regulate costochondral growth plate chondrocytes via mechanisms that are specific to the cell maturation zone from which the cells were isolated. 1α,25(OH)_2_D_3_ regulates GC chondrocytes both through the nuclear vitamin D receptor (VDR) and through a membrane-associated receptor, protein disulfide isomerase A3 (PDIA3), which mediates its effects via a protein kinase C (PKC) signal transduction pathway. PKC-alpha (PKCα) is increased via a phosphatidylinositol-specific phospholipase C (PLC)-dependent mechanism, as well as through the stimulation of phospholipase A2 (PLA2) activity. Arachidonic acid and its downstream metabolite prostaglandin E2 (PGE2) also modulate cell response to 1α,25(OH)_2_D_3_. In contrast, 24R,25(OH)_2_D_3_ exerts its effects on RC chondrocytes through a separate, membrane-associated receptor that also involves PKC pathways. However, PKCα is increased via a phospholipase D (PLD)-mediated mechanism, as well as through inhibition of the PLA2 pathway [60,61].

Matrix vesicle production is also regulated in a zone-specific manner. Studies using matrix vesicles produced by RC and GC chondrocytes in culture show that 24R,25(OH)_2_D_3_ stimulates the production of RC matrix vesicles by RC cells, including the incorporation of neutral MMPs and increased TNAP. However, it does not affect matrix vesicle production by GC cells. In contrast, 1α,25(OH)_2_D_3_ increases the activity of acid MMP and TNAP in matrix vesicles produced by GC chondrocytes, but it does not modify the RC matrix vesicle composition. These observations have been confirmed in vivo using vitamin D deficient/phosphate deficient rats as a model. The rachitic animals were treated with 25(OH)D_3_, 1α,25(OH)_2_D_3_, or 24R,25(OH)_2_D_3_; at harvest, matrix vesicles isolated from the growth plate were examined, and their MMP activities were compared. The results showed that the composition of the extracellular microsomes was comparable to those obtained using matrix vesicles isolated from cell culture [62].

Additional evidence that matrix vesicle composition is controlled genomically involves the incorporation of PKCα and PKC-zeta (PKCζ) following treatment with the vitamin D metabolites. After 24 h of exposure to the hormones, plasma membranes isolated from cell lysates contained primarily PKCα, but matrix vesicles contained primarily PKCζ. This differential enrichment of PKCζ is regulated by 24R,25(OH)_2_D_3_ in RC matrix vesicles and by 1α,25(OH)_2_D_3_ in GC matrix vesicles [48].

Matrix vesicles are also regulated once in the matrix via non-genomic mechanisms. Chondrocytes possess the ability to generate vitamin D metabolites locally [63]. Moreover, they secrete them at high levels and under regulation by hormones and growth factors [63]. In vitro, treatment of isolated naïve matrix vesicles with 1α,25(OH)_2_D_3_ or 24R,25(OH)_2_D_3_ produced an intriguing finding. Whereas 24R,25(OH)_2_D_3_ stimulated TNAP and PKCα, it inhibited PLA2 and PKCζ activity in matrix vesicles produced by RC cells [13,64,65]. In contrast, 1α,25(OH)_2_D_3_ stimulated TNAP, PKCα, and PLA2 and inhibited PKCζ in matrix vesicles from GC cells. Moreover, 1α,25(OH)_2_D_3_ activated PKCα and PLA2 in RC matrix vesicles, but its effect was much less robust than in GC matrix vesicles.

While the mechanisms by which the vitamin D metabolites act on matrix vesicles once in the matrix are not yet known, the biological consequences are becoming understood. It is evident that the primary function of 24R,25(OH)_2_D_3_ is to ensure that matrix vesicles remain structurally stable in the matrix, at least with respect to the integrity of the membrane. 24R,25(OH)_2_D_3_ treatment of RC cells results in the production of matrix vesicles with higher levels of phosphatidylcholine than GC matrix vesicles [55]. Moreover, 24R,25(OH)_2_D_3_ inhibits PLA2, thereby limiting the production of lysophospholipids, which has the effect of stabilizing the phospholipid bilayer. 1α,25(OH)_2_D_3_ treatment of GC cells results in matrix vesicles with a higher percentage of acidic phospholipids, and when GC matrix vesicles are treated directly with the hormone, PLA2 is stimulated, and production of lysphophospholipids is increased. The lysophospholipids stimulate PKCα activity [66], and they destabilize the membrane, contributing to the leakiness associated with uptake of Ca^++^ and initial mineral formation.

The destabilized membrane also provides a pathway for the release of MMPs and aggrecanases into the ECM (Figure 2). One outcome of this is activation of latent transforming growth factor-beta (TGFβ), which is stored in the ECM via latent TGFβ-binding protein [67]. The matrix vesicles lack collagenase [36], so the release of the enzymes contributes to the degradation of the proteoglycan content, thereby removing a major inhibitor of crystal growth while retaining the collagen matrix to support the calcification process [68,69]. The importance of these enzymes was demonstrated clearly when matrix vesicles were incubated in a proteoglycan-rich gel containing Ca^++^ and free phosphate [45]. Only when they were pretreated with 1α,25(OH)_2_D_3_ did they contribute to apatite formation in the gel, indicating that the release of the enzymes and subsequent degradation of glycosaminoglycan was necessary.

## 5. Matrix Vesicles as ECM-Linked Exosomes

Are MVs a species of exosome? Growth plate chondrocytes do produce vesicles typical of exosomes that they release into the culture media [9]. However, they differ significantly from matrix vesicles. They are not enriched in alkaline phosphatase activity, their phospholipid content mimics that of the plasma membrane, and they possess different enzymes than are found in matrix vesicles. Whether they are exosomes is not yet resolved, although we have reported the presence of microRNA’s [70], as have others [71,72,73].

Matrix vesicles share physical characteristics with exosomes in terms of size, but unlike exosomes, they are anchored in the ECM. As noted above, they are structurally and compositionally distinct from the media vesicles, but are they a unique species of exosomes that possess small RNAs capable of modulating cell behavior? This is an attractive idea for a number of reasons. The growth plate is not vascularized, and the chondrocytes are surrounded by a large ECM, limiting communication among cells. The arrangement of cells in the growth plate is linear, with RC cells at the top destined to undergo maturation as GC cells. Studies show that treatment of RC chondrocytes with 24R,25(OH)_2_D_3_ ultimately induces a transition to a GC phenotype, complete with a shift in response from 24R,25(OH)_2_D_3_ to 1α,25(OH)_2_D_3_ [74]. Under normal bone growth, RC cells undergo proliferation, and post-proliferative prehypterophic chondrocytes become hypertrophic, producing a new set of matrix vesicles and undertaking a massive reorganization of the ECM to accommodate their size as well as facilitate neovascularization and calcification. This occurs under mechanical load, and cells communicate with each other via the release of a variety of factors, including cytokines, growth factors, hormones, and other small molecules. Recent data suggest that matrix vesicles may contribute to this communication in a manner that is similar to exosomes.

To test this hypothesis, we undertook a careful analysis of matrix vesicle RNA content. Initial studies demonstrated clearly that a subset of microRNAs was differentially concentrated in the MVs isolated from GC chondrocyte cultures compared to the parent cells [70]. (Figure 3A,B) Similarly, four microRNA were unique to RC matrix vesicles compared to their parent cells: miR-451-5p, miR-223-3p, miR-142-3p, and miR-122-5p; and miR-22-3p was among the most abundant miRNA. Moreover, when we compared RNAs in MVs isolated from GC cultures to those isolated from RC cultures, we found distinct cell-specific differences in microRNA content [59], suggesting that they may play a differential role in chondrocyte regulation.

Experiments assessing the chondrocyte response to specific microRNAs that are selectively exported in matrix vesicles by GC chondrocytes indicate that they do have direct effects on the cells [38]. To test this, RC and GC chondrocytes were transfected with synthetic mimics of MV microRNAs. miR-122 drove both RC and GC cells toward a proliferative phenotype, stabilized the matrix, and inhibited terminal differentiation. Other MV microRNA also had a regulatory effect on the growth plate chondrocytes. miR-22 increased PTHrP and Ihh production in RC and GC chondrocytes. miR-451 decreased alkaline phosphatase activity (an important factor in tissue mineralization) in both RC and GC cells though to a lesser extent than miR-122. In addition, miR-451 increased OPG production in GC cells [38].

The MV microRNA has been reported to be present in exosomes produced by other cell types. As noted for cancer cells, miR-122 stimulates proliferation [75]. In growth plate cells, this has the effect of increasing cell proliferation, decreasing the production of maturation markers, and producing matrix components characteristic of proliferating cartilage [38]. Transfecting miR-122 into rat articular chondrocyte cells in an in vitro model of osteoarthritis results in increased proliferation and inhibition of inflammatory markers. In contrast, miR-451 stimulates the production of matrix processing enzymes, such as MMP13, which leads to remodeling of the ECM in preparation for calcification [76,77]. Together with the observation that rapid signaling at the MV membrane by vitamin D metabolites can modulate the rate and extent of release of factors from matrix vesicles as well as activate factors such as TGFβ1 and TGFβ2 in the ECM, these new data strongly support the idea that matrix vesicles act as a method to control events in the ECM as well as how cells in the growth plate undergo terminal differentiation [48,65,78,79].

## 6. Conclusions

Lessons learned from the study of MVs over the last fifty years have shown that they perform multiple functions in ECM-rich tissues such as cartilage and bone. Our early understanding that they provide sites of initial mineral formation was first based on microscopic studies, and biochemical analysis supported the concept that this was their primary role. As technology has progressed, it has become possible to examine other aspects of MV biology, showing us that they can function in an exosome-like manner, carrying microRNAs into the ECM and facilitating the dissemination of information. While this is similar to exosomes, it is distinctly different, underscoring the need to examine the role of these extracellular organelles in the context of the cells and tissues that produce them.

## Figures and Tables

**Figure 1 cells-11-01619-f001:**
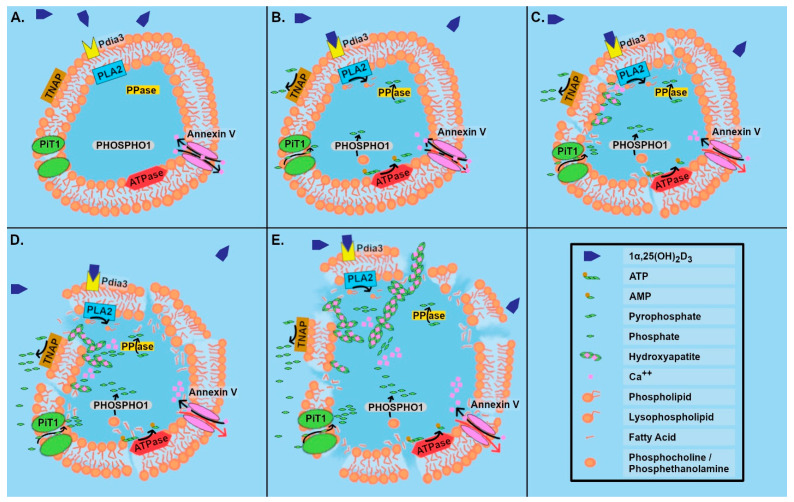
Growth zone matrix vesicle breakdown and hydroxyapatite crystal formation in response to 1α,25(OH)_2_D_3_. (**A**) Annexin V mediated transport of Ca^++^ maintains homeostasis inside vesicle. (**B**) 1α,25(OH)_2_D_3_ binding with PDIA3 on vesicle membrane activates phospholipase A2 (PLA2). This results in release of arachidonic acid and the production of lysophospholipid. ATPase activity is reduced due to lack of an energy source, so active transport of Ca^++^ out of the vesicle via annexin V is reduced. The action of ATPase produces AMP and pyrophosphate, which is a calcification inhibitor. PPase breaks pyrophosphate down into phosphates, and on the outer leaflet of the membrane, TNAP generates free phosphate that is available for transport into the vesicle by PiT1. Inside the vesicle, PHOSPHO1 releases phosphate from phosphocholine or phosphoethanolamine. (**C**) The matrix vesicle membrane becomes leaky. The first hydroxyapatite crystals have formed on the inside of the matrix vesicle following nucleation via Ca-phosphatidylserine-phosphate complexes; active transport of Ca^++^ via annexin V no longer occurs. (**D**) There is increased accumulation of calcium and phosphate along the surface of the membrane. (**E**) Membrane integrity is lost, and hydroxyapatite crystals grow out into the ECM.

**Figure 2 cells-11-01619-f002:**
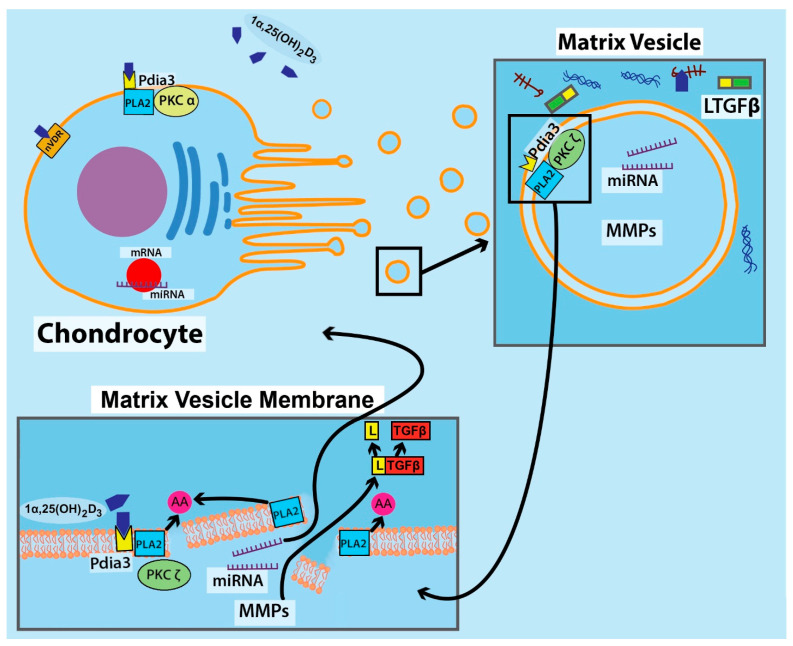
Growth zone chondrocyte with plasma membrane-associated receptors (VDR and PDIA3) for 1α,25(OH)_2_D_3_ release matrix vesicles into the ECM that contain microRNA and matrix metalloproteinases (MMPs). Chondrocyte plasma membrane PDIA3 acts via PKCα. Matrix vesicles contain the PDIA3 receptor complexed with PKCζ. 1α,25(OH)_2_D_3_ binds to matrix vesicle PDIA3, activating PLA2 to release arachidonic acid and destabilize the vesicle membrane; this releases the contents into the ECM. MMP released from MVs activates latent TGFβ (LTGFβ) in the extracellular matrix removing the latent binding protein. MicroRNAs act back on the chondrocytes, but whether the microRNAs are released into the ECM or are transported to the cells via the matrix vesicles or as membrane liposomes is not known.

**Figure 3 cells-11-01619-f003:**
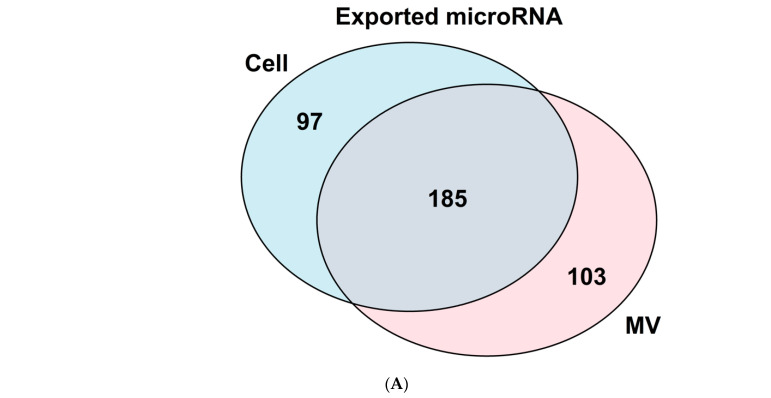
Examination of chondrocyte and matrix vesicle microRNA populations. (**A**) Venn diagram comparing differentially expressed (*p*-value < 0.05 and absolute log 2-fold change > 1) microRNA found in growth zone chondrocytes and matrix vesicles with 97 microRNA found only in the chondrocytes, 103 only in vesicles, and 185 shared between cells and vesicles. (**B**) Heatmap of the differentially expressed (*p*-value < 0.05 and absolute log 2-fold change > 1) microRNA from growth zone chondrocytes with cells on the left and matrix vesicles on the right (*n* = 3).

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
