# Peer review of "The Role of Matrix-Bound Extracellular Vesicles in the Regulation of Endochondral Bone Formation"

_cells, 2022, doi:10.3390/cells11101619_

Round 1
Reviewer 1 Report
In this review by Boyan B.D. et al., authors provide a detailed overview of the role of matrix vesicles in the process of endochondral ossification. In particular, the authors provide their expertise to extensively describe the biogenesis and function of matrix vesicles.
Major Points
1) The review identifies a niche although, to be more relevant and of broad interest to the scientific community, it would be advisable to structure more clearly this current format to highlight the evidences collected.
2) To improve the writing for a better flow, it would be very useful to the reader if the author could provide a small introduction listing the main points. The first three chapters, in fact, contain redundant information which can appear slightly out of topic with respect to the special issue "Extracellular Vesicle-Associated Non-Coding RNAs”, so I suggest to revise and shorten a bit this part. In this regard, the details on their mechanism of formation of matrix vesicles are much more described than the potential role of this extracellular RNA, prevalently highlighted in the Chapter 4.
3) The authors sometimes refer to these matrix vesicles in the text as exosomes or microvesicles. Although the authors explain the unique nature of these vesicles, it would be useful to the community to discuss more clearly this point in a single paragraph at the beginning. It is not always clear to the reader if the use of this nomenclature refers to size or to a partial shared biogenesis. Moreover, it is advisable to collect the technical details (e.g., the isolation method) in a single paragraph rather than in multiple sections (lines 66-69; 86-92; 157-167; 172-176) and to associate this information along with the type of bioactive molecules they carry, helping to contextualize better these vesicles in the field.
4) I have found Figure 3 not appealing as the author do not provide enough methodological background to interpret the data (lines 297-298). I would rather suggest to construct a table to list and highlight the RNA species associated to matrix vesicles, describing their biological function and potential perspective clinical applications. This type of graphical support would be more immediate to facilitate the readers’ understanding.
5) In the final paragraph, I would expect the authors to elaborate a few conclusions on the evidences of the RNAs found in matrix vesicles in parallel with the current literature on extracellular vesicles, to separate this work from many publications that refer to the same RNA species detected in a wide range of vesicles. Additionally, if the author could provide their vision about the clinical potential of the RNAs shuttled by these vesicles, it would add a value to the review and to the scientific community.
Minor points
Some passages are very long (e.g. 103-109), some typo must be fixed (Latin letters not appearing), a few name of molecules are mentioned first as acronyms and only later in their extended version. I suggest to revise the references as some citations appear redundant (e.g., Ref. 12 and 13).
Author Response
Manuscript #1660933
Title: “The Role of Matrix-Bound Extracellular Vesicles in the Regulation of Endochondral Bone Formation”
Authors: Boyan et al.
Response to Reviewer 1
In this review by Boyan B.D. et al., authors provide a detailed overview of the role of matrix vesicles in the process of endochondral ossification. In particular, the authors provide their expertise to extensively describe the biogenesis and function of matrix vesicles.
Major Points
1) The review identifies a niche although, to be more relevant and of broad interest to the scientific community, it would be advisable to structure more clearly this current format to highlight the evidences collected.
The review has been restructured to make content and goals clearer. An introduction was added that describes matrix vesicles (their discovery, characteristics, known functions, regulation and the procedure used to isolate them) and lays out the goals of this review – to examine their role in cartilage tissue, investigate if they are a subclass of exosomes, and discuss recently discovered and characterized microRNA within these matrix vesicles. Aspects of matrix vesicles that are integral to their activity within the growth plate (being bound to collagen and the lack of tissue vascularization) provide insight into the value of the growth plate as a system to study these vesicles and some of the defining characteristics of matrix vesicles.
2) To improve the writing for a better flow, it would be very useful to the reader if the author could provide a small introduction listing the main points. The first three chapters, in fact, contain redundant information which can appear slightly out of topic with respect to the special issue "Extracellular Vesicle-Associated Non-Coding RNAs”, so I suggest to revise and shorten a bit this part. In this regard, the details on their mechanism of formation of matrix vesicles are much more described than the potential role of this extracellular RNA, prevalently highlighted in the Chapter 4.
Thank you for this suggestion, the review paper has been reworked, according to your suggestion, to have the following chapters: an introduction detailing MVs and the goals of this review, an overview of the growth plate and how it is structured and regulated, a detailed description of matrix vesicles and how they are distinct to growth plate tissue while having variation within the growth plate itself, the primary regulators of matrix vesicles, followed by a discussion of whether or not matrix vesicles are a class of matrix bound exosomes, and finally a conclusion. We agree that this provides a better flow for the reader about the tissue and vesicles involved and why it remains an open question as to the exact classification of matrix vesicles especially with the recent discovery that they contain microRNA capable of regulating the chondrocytes in the growth plate.
3) The authors sometimes refer to these matrix vesicles in the text as exosomes or microvesicles. Although the authors explain the unique nature of these vesicles, it would be useful to the community to discuss more clearly this point in a single paragraph at the beginning. It is not always clear to the reader if the use of this nomenclature refers to size or to a partial shared biogenesis. Moreover, it is advisable to collect the technical details (e.g., the isolation method) in a single paragraph rather than in multiple sections (lines 66-69; 86-92; 157-167; 172-176) and to associate this information along with the type of bioactive molecules they carry, helping to contextualize better these vesicles in the field.
Thanks for your comment. We went through the text to review the nomenclature and have dropped the term ‘microvesicles’ from the text. The review has been reworked to make it clear what characteristics define matrix vesicles and what their currently accepted, primary role is within the growth plate (centers of mineralization). With the recent discovery of microRNA capable of regulating chondrocytes within the growth plate, additional roles have been opened up for these vesicles and the question of whether they are distinct from exosomes or a subclass of them that remain bound to the matrix of the growth plate is discussed.
The isolation methods and approaches have also been consolidated to tissue and cell culture examinations and the bioactive molecules to the chapter detailing matrix vesicle function.
4) I have found Figure 3 not appealing as the author do not provide enough methodological background to interpret the data (lines 297-298). I would rather suggest to construct a table to list and highlight the RNA species associated to matrix vesicles, describing their biological function and potential perspective clinical applications. This type of graphical support would be more immediate to facilitate the readers’ understanding.
The first panel of figure 3 (A) has been removed and both the text and legend reworked to clarify the Venn diagram and heatmap. As the primary goal of the figure is to detail the distinct populations of microRNA found in matrix vesicles compared to the cells that produce them, we believe that the figure provides this information to the reader. The Venn diagram is there to show that the vesicles contain a set of microRNAs not detectable in the chondrocytes and therefor produced and selectively exported in matrix vesicles. The heatmap is visualizing the differentially expressed microRNA populations and demonstrating that they are distinct between the cells and matrix vesicles. The paper then goes on to detail the research that has been done on specific microRNA and their impact on both articular and growth plate chondrocytes.
5) In the final paragraph, I would expect the authors to elaborate a few conclusions on the evidences of the RNAs found in matrix vesicles in parallel with the current literature on extracellular vesicles, to separate this work from many publications that refer to the same RNA species detected in a wide range of vesicles. Additionally, if the author could provide their vision about the clinical potential of the RNAs shuttled by these vesicles, it would add a value to the review and to the scientific community.
We have added information about the selectively exported microRNA and their effect on chondrocytes in vitro. This information details the ability of these microRNA to regulate chondrocytes in both the growth plate and articular cartilage. It focuses on the effects observed in cell culture conditions in chondrocytes from the growth plate and then how articular chondrocytes respond to these microRNA. This takes the work from the growth plate to the joint where there is a need for improved clinical interventions to treat arthritis.
Minor points
Some passages are very long (e.g. 103-109), some typo must be fixed (Latin letters not appearing), a few name of molecules are mentioned first as acronyms and only later in their extended version. I suggest to revise the references as some citations appear redundant (e.g., Ref. 12 and 13).
The font and Latin letters have been double checked and are all in arial and should display correctly when uploaded. The acronyms have also all been reviewed and the list at the beginning of the paper updated to reflect that. Citations have also been reviewed and made more extensive.

Reviewer 2 Report
Review for “The Role of Matrix-Bound Extracellular Vesicles in the Regulation of Endochondral Bone Formation”
In the review for “The Role of Matrix-Bound Extracellular Vesicles in the Regulation of Endochondral Bone Formation” the authors give an overview of the role and function of matrix vesicles in the process of mineralization. This review is very well written and takes advantage of the current research interest in extracellular vesicles and the authors experience to talk about matrix vesicles biogenesis, function, and microRNA transportation. Overall, the article is of significant interest to the field with some issues that need to be addressed, especially in the references section.
Major comments:
The limited number of citations – For a field of research that has been around for decades, it is surprising why there are so few references. Can the authors provide reasoning they have opted to use only 61 references?
The high percentage of self-citations – The first author is part of more than 1/3 of the references. While the references are properly used a better overview of other authors' work within the field would be beneficial.
Lack of recent publications – Only 14 of the publications include data from the last 10 years. More recent articles should be included
Figure 1 is difficult to interpret and analyze – Overall the figure is not intuitive and difficult to interpret without supporting text. For instance, AMP/ATP, Phosphate/Pyrophosphate are too similar and very difficult to distinguish even with supporting legend. Moreover, the number of panels (A,B…E) could be shortened to 2 or 3 panels for ease of interpretation
Is the summary necessary? – Could the authors explain why they opted for doing a summary instead of a conclusion? Abstract should state a similar message to a summary. The document finishes with “It is clear that rapid responses to steroid hormones, like 1α,25(OH)2D3 and 24R,25(OH)2D3, provide a well-regulated mechanistic approach for modulating the rate and extent of microRNA release”. The document would significantly improve if the authors could discuss a bit further this topic
Minor comments:
Choice of keywords – Choice of keywords seem odd. The majority of the words are not included in the abstract and only appear at later stages of the review. For instance, do the authors consider the protein kinase C more relevant than microRNA?
The authors used several times the following expression “right-side” and “wrong-side”. Can they explain and clarify what they mean by it?
On lines 227 to 230, the PKC word is likely missing an extra symbol. The same was observed with other words like TGF.
Overall, the document is sound and definitely worth to be published once the issues described have been addressed.
Author Response
Manuscript #1660933
Title: “The Role of Matrix-Bound Extracellular Vesicles in the Regulation of Endochondral Bone Formation”
Authors: Boyan et al.
Response to Reviewer 2
In the review for “The Role of Matrix-Bound Extracellular Vesicles in the Regulation of Endochondral Bone Formation” the authors give an overview of the role and function of matrix vesicles in the process of mineralization. This review is very well written and takes advantage of the current research interest in extracellular vesicles and the authors experience to talk about matrix vesicles biogenesis, function, and microRNA transportation. Overall, the article is of significant interest to the field with some issues that need to be addressed, especially in the references section.
Major comments:
The limited number of citations – For a field of research that has been around for decades, it is surprising why there are so few references. Can the authors provide reasoning they have opted to use only 61 references?
Citations have been reviewed and expanded. There are now 81 references for the review paper.
The high percentage of self-citations – The first author is part of more than 1/3 of the references. While the references are properly used a better overview of other authors' work within the field would be beneficial.
Dr. Boyan and Dr. Schwartz have been focused on matrix vesicles and this field for a number of decades and have led much of the research into this subject. The expanded reference list does not change the percent of citations involving Dr. Boyan, it does provide more background into the content.
Lack of recent publications – Only 14 of the publications include data from the last 10 years. More recent articles should be included
There are now 20 publications cited from the past 10 years.
Figure 1 is difficult to interpret and analyze – Overall the figure is not intuitive and difficult to interpret without supporting text. For instance, AMP/ATP, Phosphate/Pyrophosphate are too similar and very difficult to distinguish even with supporting legend. Moreover, the number of panels (A,B…E) could be shortened to 2 or 3 panels for ease of interpretation
We made some edits to the text and figure legend. The figure attempts to closely follow the paragraph detailing matrix vesicle mineralization in the growth plate. This is a key characteristic of matrix vesicles and one of their defining roles. The figure is working to demonstrate the initiation of this process following activation by 1α,25(OH)2D3 in growth zone matrix vesicles. The numerous panels are depicting the gradual process of crystal formation and matrix vesicle breakdown. The bulk of the description takes place for panels A & B and panels C through E visualize the process while keeping it from being overwhelming.
Is the summary necessary? – Could the authors explain why they opted for doing a summary instead of a conclusion? Abstract should state a similar message to a summary. The document finishes with “It is clear that rapid responses to steroid hormones, like 1α,25(OH)2D3 and 24R,25(OH)2D3, provide a well-regulated mechanistic approach for modulating the rate and extent of microRNA release”. The document would significantly improve if the authors could discuss a bit further this topic
Thanks for your feedback, we have removed the summary section and there is now a conclusion. The structure of the paper has also been reworked. An introduction has been added and a clear focus for the review paper has been included. The review provides a detailed background on matrix vesicles discussing both their similarity to and difference from traditional exosomes. The recent discovery and examination of microRNA in the matrix vesicles adds to this discussion.
Minor comments:
Choice of keywords – Choice of keywords seem odd. The majority of the words are not included in the abstract and only appear at later stages of the review. For instance, do the authors consider the protein kinase C more relevant than microRNA?
The keywords have been updated and the order changed. The vitamin D metabolites are discussed throughout the paper and key regulators of the chondrocytes and matrix vesicles. PKC was dropped and microRNA added to the list.
The authors used several times the following expression “right-side” and “wrong-side”. Can they explain and clarify what they mean by it?
This section of the text was reworded to better explain what is meant and the term ‘wrong-side’ was dropped from the text all together.
“Because the MVs are not lysed during isolation, their native conformation remains intact; thus, they remain right side out, with the ectoenzyme TNAP facing outward. In contrast, PMs are isolated from lysed cells and may form liposomes with the outer leaflet of the membrane facing the interior, away from the TNAP substrate in the media.”
On lines 227 to 230, the PKC word is likely missing an extra symbol. The same was observed with other words like TGF.
The document has been checked and all characters are in arial font and should be able to be processed by the publisher without incident. The symbols on line 227 to 230 happened after document upload and review during one of the processing steps.
Overall, the document is sound and definitely worth to be published once the issues described have been addressed.

Round 2
Reviewer 1 Report
I thank the authors for the time spent to critically address my comments. The manuscript has improved and I believe that most of my comments have been agreeably addressed without requiring further revisions. I would only suggest a final spell check as I still have spotted a few typo along the text.
